# Reaching a Wider Audience: Instagram’s Role in Dairy Cow Nutrition Education and Engagement

**DOI:** 10.3390/ani13223503

**Published:** 2023-11-13

**Authors:** Edlira Muca, Giovanni Buonaiuto, Martina Lamanna, Simone Silvestrelli, Francesca Ghiaccio, Alessia Federiconi, Julio De Matos Vettori, Riccardo Colleluori, Isa Fusaro, Federica Raspa, Emanuela Valle, Andrea Formigoni, Damiano Cavallini

**Affiliations:** 1Department of Veterinary Sciences, University of Turin, 10095 Grugliasco, Italy; federica.raspa@unito.it (F.R.); emanuela.valle@unito.it (E.V.); 2Department of Veterinary Sciences, University of Bologna, 40064 Bologna, Italy; giovanni.buonaiuto@unibo.it (G.B.); martina.lamanna2@studio.unibo.it (M.L.); simone.silvestrelli2@unibo.it (S.S.); francesca.ghiaccio2@unibo.it (F.G.); alessia.federiconi3@unibo.it (A.F.); riccardo.colleluori2@unibo.it (R.C.); andrea.formigoni@unibo.it (A.F.); 3Faculty of Veterinary Sciences, University of Teramo, 64100 Teramo, Italy; jdematosvettori@unite.it (J.D.M.V.); ifusaro@unite.it (I.F.)

**Keywords:** dairy nutrition, Instagram education, veterinary social media, digital outreach

## Abstract

**Simple Summary:**

Despite its significance, the general public frequently misses out on scientific knowledge about the nutrition and health of dairy cows. Our study used Instagram, a well-known social media platform, to engage and inform a larger audience in order to solve this issue. Making dairy cow nutrition and welfare understandable and pertinent to common people was our aim. We created our Instagram account, where we posted a variety of content, including research insights and the firsthand accounts of our students’ time spent working on our dairy farm. The outcomes were encouraging: we created a community that prioritizes the welfare of dairy cows, and our followers learned more about the relationship between cow health and the quality of dairy products.

**Abstract:**

The paper presents an overview of the usage of Instagram as a social media platform for teaching and engagement in the field of dairy cow nutrition and management. Our Instagram content, which includes posts, stories, hashtags, reels, and live videos, aims to educate and engage our followers and covers a wide range of topics, including research updates, student activities, and information on dairy cow health and welfare. This approach to education allows us to reach a larger audience while also providing a forum for interaction and discussion on essential dairy cow nutrition concerns.

## 1. Introduction

Social media platforms have substantially advanced in recent years from being merely communication channels to becoming effective tools for knowledge sharing, promoting learning, and creating professional networks [1]. Instagram, which is recognized for its aesthetic style, distinguishes among these platforms related to its broad audience, particularly among younger users [2,3]. Existing research suggests that Instagram is being used as an educational tool in higher education [4]. Instagram has been found to facilitate self-directed learning and peer mentoring among university students, with positive attitudes and a strong focus on educational goals [5]. It has also been utilized in medical education to share image-based information, helping to develop clinical reasoning and critical thinking skills [3,6,7]. Additionally, Instagram has been used in dental education to create collaborative online learning spaces, resulting in meaningful learning experiences and the application of theoretical concepts to professional practice [8].

There is a rising demand in the field of dairy cow nutrition and management to convert difficult scientific concepts into comprehensible knowledge. This need extends beyond the traditional classroom. It reaches to the general population, who are becoming more concerned with how their food is produced and the welfare of the animals involved. Recognizing the gaps in the current state of knowledge regarding the use of social media to promote specialized, scientific agricultural knowledge [9], we embarked on this journey. Our main goal was to establish an interactive learning platform that engages both students and professionals in the field, as well as the general public, in understanding the dairy cow nutrition and management. Instagram’s features allow for the creation of a virtual learning community where ideas can be exchanged, and knowledge can be transferred. This short communication explores our journey in creating and managing an Instagram account dedicated to dairy cow nutrition and management. We outline our strategy for appropriate content creation, audience engagement, and the use of Instagram features. Insights gained from our interaction with followers and the analytics provided by the platform are shared, highlighting the potential of Instagram as an educational tool. Our dual approach aimed to enhance learning outcomes and build a community with common interest in dairy cow nutrition while providing individuals with revised information and accurate knowledge of the field. We acknowledge that the present contribution will inspire other educators to employ the potential of social media platforms, in particular Instagram, for teaching and engagement in the current digital era.

## 2. Setting Up an Instagram Account for Our Dairy Cow Nutrition University Farm

We created our Instagram account (stalladidattica_unibo) in December 2020. This initiative aimed to modernize our department, bridging the gap between nutrition and management practices and veterinary students, practitioners, and enthusiasts. Our primary objectives were to emphasize the significance of proper nutrition and management as well as to share important department-related events and updates. Several successful strategies were implemented within the content creation and engagement phase. For example, the “Know Your Feed” series highlighted various types of feed, their nutritional characteristics, and the recommended feeding procedures, resulting in a significant boost in engagement rates. Furthermore, our weekly question-and-answer sessions evolved into a forum for direct engagement, allowing us to create trust and connection with our audience. Among the primary methods that increased our awareness and engagement were regular posting, interactive content such as polls and quizzes, and collaborations with similar accounts. Figure 1 provides an overview of our Instagram page, showcasing the different features and content we offer. Key elements include the following:Bio: Describes our department’s vision, goals and expertise.Stories: Provides simple information on dairy cow nutrition and management.Direct messaging: Facilitates discussions and answering questions.Contact info: Included in the bio for collaborations and resource sharing.Highlighted stories: Selected stories lasting beyond the usual 24-h timeframe, creating a content repository.Post feed: Provides a variety of educational content related to dairy cow nutrition and management.IGTV: Shares in-depth video content for an all-around learning experience.Tagged posts: Promotes community interaction by recognizing and reposting tagged posts.Reels: Delivers short videos on a wide range of instructional topics.

We thus established a dynamic platform that enriches veterinary education in the field of dairy cow nutrition and management by leveraging these features while providing relevant educational information.

## 3. Leveraging Instagram as a Robust Teaching and Learning Tool in Veterinary Education and Practice: Lessons from a Dairy Cow Nutrition and Management Page

Given the growing preference for Instagram among medical students as a primary social media platform for educational content [10,11], veterinary educators, researchers, and practitioners must consider the potential of this platform as an innovative educational tool. As the social media landscape evolves among young veterinarians [12], veterinary sciences can benefit from Instagram, which has proven to be particularly attractive to the younger generation, with 90% of its 150 million users under 35 years of age [11]. As a result, incorporating Instagram as a teaching and learning tool effectively can result in a significant platform for knowledge sharing, collaboration, and engagement among veterinary professionals and students. Setting clear objectives, creating a professional account for access to insights, creating relevant and engaging content, ensuring quality and consistency, and fostering interaction through Instagram features are all necessary for successful Instagram integration in veterinary education. This strategy is illustrated on the Dairy Cow Nutrition and Management page. For each stage, a specific explanation is provided in subsequent sections to help detail the successful implementation of these techniques.

## 4. Establish Clear Objectives

Establishing clear objectives is paramount for creating an effective and engaging educational Instagram page. The specific objectives will vary based on the topic and area of specialty [3,13]. To establish clear goals for our Instagram feed, we first identified the major difficulties and myths in dairy cow nutrition and management. We were able to adjust our aims to address these specific challenges by analyzing the prevalent gaps in knowledge and disinformation spreading on the internet. We also evaluated the audience we intended to target, ranging from students to experts, and made certain that our goals reflected their various degrees of competence. Regular input from our followers as well as periodic evaluations of our content’s engagement analytics helped us modify and realign our goals to ensure they remained relevant. Our dairy cow nutrition page, for example, aims to educate veterinary students, animal science students, practitioners, farmers, and other enthusiasts about the most recent research and best practices in optimizing dairy cow nutrition as well as to counteract misinformation prevalent on the internet concerning this field. The cognitive goal of our initiative was to effectively disseminate current scientific research and knowledge in dairy cow nutrition and management. The utilitarian goal was to provide practical, actionable insights that can be immediately applied by practitioners in the field. This was put into practice through a campaign that highlighted sustainable feeding practices, which included infographics, video interviews with experts, and interactive Q&A sessions. By addressing the knowledge gap, our study aimed to contribute to the limited body of research on the use of social media for disseminating specialized agricultural knowledge, particularly within the context of Instagram’s unique multimedia capabilities. Another important goal was to share accurate information. We hoped to offset the spread of incorrect and fake information on dairy cow nutrition on the internet by sharing scientific knowledge and eliminating myths. Furthermore, the page aims to provide non-experts with access to information relating to this field, allowing for an increase in the interest of the population in this area. Finally, we highlight best practices in dairy cow nutrition and management by displaying successful cases from our university dairy farm and well-managed dairy operations. We created particular assessment criteria to guarantee that these objectives were not only clearly defined but also quantifiable. Quantitative measures such as engagement rates (likes, comments, and shares), follower growth, and post reach give real evidence of our page’s popularity and the resonance of our content with our target audience. We use a qualitative form of feedback via direct messages and comments to assess the audience’s comprehension and implementation of the knowledge provided. Periodic surveys will also be undertaken in the future to obtain comprehensive feedback on the content’s utility and recommendations for improvement.

## 5. Professional Account Setup and Utilization of Instagram Insights

To gain access to Instagram’s analytical feature, i.e., Insights, it is necessary to set up a professional account, which should be public to ensure accessibility to Insights [10,14]. A public account allows all Instagram users to discover the account, providing potential followers an opportunity to peruse its content before deciding to follow. Our experience with Instagram Insights has been an essential contributor to understanding our audience’s interaction with our content. It provides us with an in-depth look at the performance of certain posts, stories, videos, and reels, allowing us to assess their effectiveness and the level of engagement they generate. These Insights, which are free and only available to professional accounts, can be accessed via the Instagram app’s profile. It offers an overview of recent highlights, accounts reached, engagement metrics, total followers, and content shared [10,14]. It is noteworthy that Insights yields valuable demographic information about an account’s followers, including their age, gender, and geographical location. Furthermore, it provides data on the peak usage times of the app during the past week and month. These data are helpful for scheduling and customizing content to maximize follower engagement. For instance, our current Instagram following consists of 55.7% women and 44.2% men, with the 25–34-year-old age group (45.4% of the total) being the most frequent visitors to our profile, closely followed by the 18–24-year-old age group (34.6%). The most active hours for profile views are between 12 p.m. and 6 p.m. on Fridays (see Figure 2).

## 6. Create Content in Accordance with Specific Themes

Maintaining a focused Instagram feed requires that all content aligns with one’s specific topic [3,4]. For example, our Instagram page is focused on postings related to dairy cow nutrition and management. The broad array of topics we explore on our dairy cow nutrition and management Instagram page includes the chemical and nutritional characteristics of various feeds used in dairy cow diets. We share posts on laboratory feed analysis and introduce our audience to the latest precision livestock farming technologies. Posts elucidating the chemical and nutritional composition of common dairy cow feeds serve an educative purpose, helping our audience understand the components of the feeds. This knowledge is indispensable for ensuring that cows receive a balanced diet that meets their nutritional needs, promoting their health and productivity. We advocate for laboratory feed analysis in our postings to emphasize the need for high-quality, contaminant-free diets. These postings provide important insights to our readers, demonstrating how laboratory analysis helps to ensure the safety and nutritional value of the feed. By demonstrating our dedication to quality and safety, we can establish ourselves as a reliable information source, creating a devoted audience. Promoting laboratory feed analysis via social media posts is an excellent technique for endorsing safe, high-quality feeds and developing our relationship with our audience, especially as students are the bulk of our audience. Our page also includes scientific articles published by our research group and information on congresses that our team has attended. Sharing scientific publications on our Instagram feed helps us to reach out to a larger audience, which includes students, veterinarians, and academics. We carefully select articles that provide practical recommendations for on-farm activities or strengthen basic understanding in the field to ensure the information is both relevant and helpful. Each scientific paper is accompanied by a summary and, if appropriate, commentary for better comprehension, ensuring that the content is accessible to a wide range of readers. Our aim in providing access to these scientific papers is to advocate for evidence-based veterinary medicine and encourage our followers to stay updated with the latest research in dairy cow nutrition and management. In addition, we include relevant published papers in the field that present complex scientific knowledge in a more digestible format, making it easier for farmers and the general public to understand this information. Attending conferences and congresses is also important for networking, staying up to date on the newest research, and sharing ideas with other professionals in the field. By publishing updates on the congresses that our team has attended, we hope to motivate our followers to attend similar events and take advantage of comparable professional development opportunities. The significant majority of our blogs concern scientific activity and publishing, emphasizing the importance of research in education and everyday life. Sharing published papers on Instagram increases team excitement and promotes continuous education. We have discovered that sharing our published articles allows our audience to acquire updated field information. Furthermore, we promote our students’ activities at the dairy farm, such as their research projects and practical experiences. This is critical for a number of reasons. For starters, it emphasizes the importance of hands-on experience in veterinary education and its role in improving the learning process. We illustrate the practical skills and knowledge our students gain at the dairy farm by sharing these experiences. Displaying student work and experiences might motivate other students who want to pursue a career in veterinary medicine, particularly in dairy cow nutrition and management. We value our students’ privacy and acquire their permission before sharing information on their behalf. Students typically write about their activities on their personal accounts and tag the farm’s page, allowing us to republish their material and highlight their essential contributions at the dairy farm. Our page offers a lot of information on cattle breeds and their characteristics. We often post this information via Instagram stories since it enables engagement with our audience. Every Tuesday, we engage our audience in interactive quizzes about cattle breeds, teaching them the distinct characteristics of each breed. We frequently discuss the environmental effect of dairy farms, with the goal of raising awareness and sharing information on reduction measures. We are devoted to encouraging sustainable and responsible agricultural methods, and some of our postings address the environmental effect of dairy farms. We believe that by sharing this knowledge, we will create a greater awareness of the environmental effect of dairy production and support the adoption of more sustainable techniques. Another major focus of our page is animal welfare, where we highlight best practices for ensuring the well-being of dairy cows, in collaboration with the Ethology and Animal Welfare research unit. We believe that by sharing this knowledge, dairy farmers will be encouraged to emphasize animal welfare in their operations, promoting ethical and sustainable dairy production [15]. Finally, we hope to keep a lighthearted tone on our Instagram profile. We frequently post humorous images of dairy cows to celebrate holidays and special occasions, for instance, sharing well-wishes with our followers for Christmas, New Year, or Easter. These posts help create a more engaging and empathic presence on social media, showing that we are not just a source of information but also a community that values fun and creativity.

## 7. Quality, Variety, and Regularity of the Posts

When developing material for our Instagram feed, we adhere to the platform’s image and video quality guidelines. We believe that effective communication and engagement require high-quality images. To measure the success of our Instagram initiative, we utilize key performance indicators (KPIs) such as engagement rate, follower growth, and content reach. To assess the effectiveness of our methods for different types of content and audiences, we regularly analyze engagement metrics for different post types. To keep things interesting, we make sure that our articles span a wide range of themes and forms, such as photographs, videos, Instagram stories, reels, and live videos. We have discovered that variety keeps our audience interested and fosters engagement and discussion. We advocate a balanced strategy to post frequency that keeps the audience interested but not overwhelmed. Our best posting frequency has been three times per week, which allows us to maintain a continuous presence while keeping our material new and relevant. Furthermore, we attempt to keep our content consistent in order to keep our audience informed and engaged. This continuous involvement has assisted us in forming a community around our Instagram profile. Our content strategy relies on visual and multimedia content, including photos, videos, and infographics, to deliver educational content in an engaging way. In addition to adhering to platform guidelines, we have internal quality control measures, including fact-checking and peer-review processes, to ensure the accuracy and quality of our posts.

## 8. Instagram Stories and Hashtags

Instagram stories and hashtags play an important role in our Instagram strategy. For instance, we use stories to share behind-the-scenes content and hashtags like #DairyFacts to boost engagement and create themed content series. We utilize Instagram stories to post daily updates, behind-the-scenes footage, and interactive material like quizzes. These stories enable us to interact with our audience in a more casual and conversational manner. Hashtags, on the other hand, are used to categorize and make our content more discoverable. We employ a range of hashtags pertaining to our field and the post’s specific content. This not only makes our postings more visible, but it also allows us to reach a larger audience. Over time, these hashtags have also helped us establish a distinct brand identity on Instagram. Some frequently used hashtags include #UniBo, #UniVet, #Dimevet, #StallaDidattica, #University, #Dairy, #DairyFarm, #Farm, #Cattle, #Cow, #Cows, #Veterinary, #Vet, #Study, #FarmsOfInstagram, and #DairyFarming. Using these hashtags help establish and maintain our brand identity and build a community around our content.

## 9. Instagram Reels and Live Videos

Instagram reels, which allows users to share short music-accompanied videos, has proven to be a highly engaging format. We utilize reels to communicate entertaining and educational material, such as farm activity and farming system demonstrations. Reels are often used to showcase short tutorials or demonstrations. Instagram live videos, on the other hand, allow us to communicate with our audience in real time. We generally go live at conferences, journal clubs, or agricultural activities, allowing us to share our experience and knowledge in real time. Our capacity to communicate with our audience and share our work has dramatically improved because of the utilization of reels and live videos.

## 10. Instagram as a Powerful Teaching and Publicizing Tool

Instagram has proven to be an effective tool for teaching and learning because of its multimedia content and high user engagement. We utilize the platform to provide a variety of instructional information that is targeted to various learning styles and interests. We emphasize user engagement by incorporating interactive content, such as user-generated content, polls, and Q&A sessions, inviting users to participate actively in conversations. Our Instagram feed is a complete resource for anybody interested in dairy cow nutrition and management, with everything from simple explanations for the general public to more advanced conversations for specialists. Furthermore, Instagram’s interactive nature allows for active interaction and the sharing of ideas, which enhances the learning experience. Instagram is an effective vehicle for publicizing our work and reaching a broader audience in addition to its significance as a teaching tool. We believe that by sharing our findings and views, we will encourage more veterinary students to consider specializing in animal nutrition, furthering the discipline. Finally, we hope to close the interest gap in veterinary nutrition by raising awareness of the importance in the field [16].

## 11. Conclusions

We have used Instagram to educate and involve people in the subject of dairy cow nutrition and management, with encouraging outcomes. We were able to attract attention to our cause and develop a community of those interested in improving the area by taking advantage of the platform’s capabilities and user base. Furthermore, using Instagram has assisted us in gaining awareness for our work and inspiring the future generation of veterinary professionals. With social media platforms’ ever-changing features and capabilities, we believe that there is a great possibility for increased integration of these technologies into veterinary education and practice. While our strategy has achieved positive outcomes, we acknowledge limitations such as social media platforms’ ever-changing algorithms and the vast amount of misinformation circulating online. We recognize that the use of Instagram as a teaching tool in other academic disciplines and communities may be limited by factors such as the need for content to be created for specific fields, different levels of digital literacy among the audience, as well as the challenge of ensuring that the content is both engaging and educational. We are always working to enhance our strategy for dealing with these difficulties. Looking forward, we seek to continually adapt our strategy to capitalize on the ever-changing characteristics of social media platforms. We also acknowledge working with other educators and researchers to further investigate the use of social media in veterinary education. The ability of Instagram to contribute to educational growth across several disciplines needs further investigation, particularly in terms of how it can be tailored to address the different requirements and challenges faced by various academic communities. We are also aware of the difficulties of determining long-term educational benefit beyond engagement indicators. Furthermore, the potential self-selection bias of our audience as well as the necessity to be accessible to a broad spectrum of knowledge levels add to the challenges. These limitations underline the importance of future studies to evaluate the long-term educational impact of Instagram, particularly in enhancing knowledge retention and practical application in diverse learning environments.

## Figures and Tables

**Figure 1 animals-13-03503-f001:**
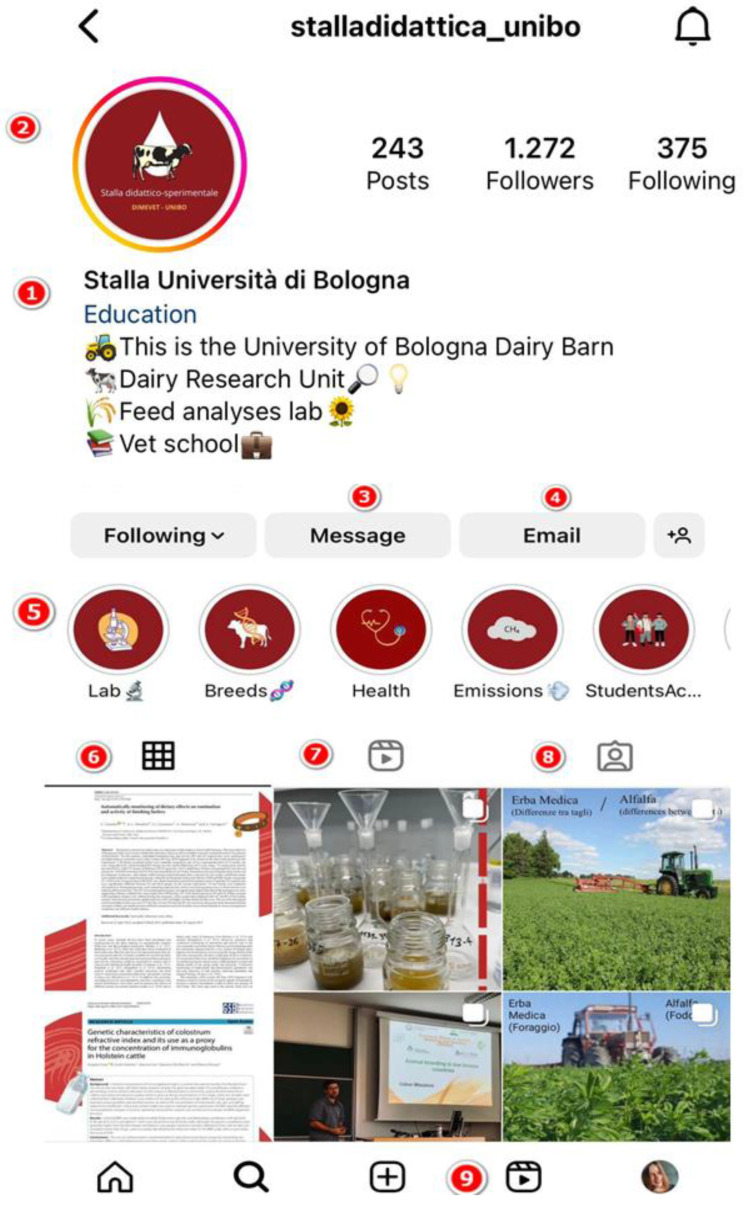
Overview of our Instagram page. (1) Instagram account and biography, (2) stories, (3) messaging, (4) email, (5) highlighted stories where those we do not want to disappear after 24 h are collected, (6) main page with all posts, (7) Instagram TV, (8) posts in which we are tagged, and (9) reels.

**Figure 2 animals-13-03503-f002:**
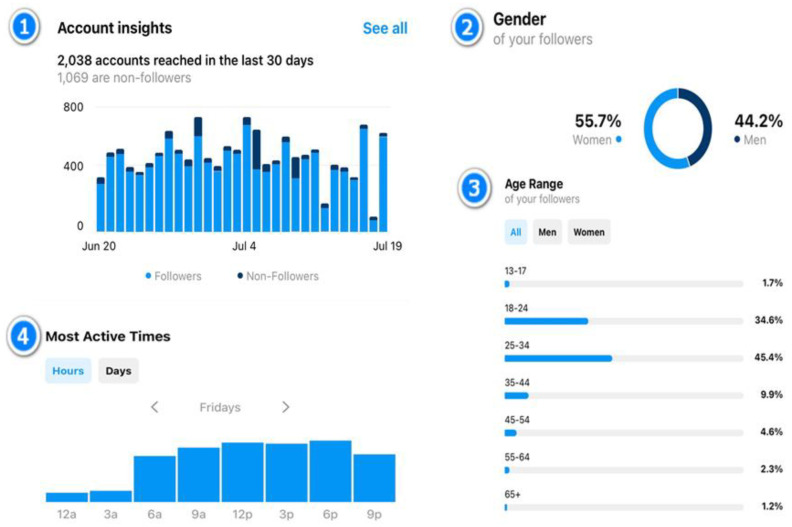
Instagram Insights is a feature available to professional accounts that allows users to analyze various metrics, including the number of visits, reach, and demographic data of followers. (**1**) The header indicating data analyzed from the past month; (**2**) a percentage graph showing the gender distribution of followers; (**3**) age and distribution data of followers; (**4**) time bands indicating peak profile-viewing hours.

## Data Availability

The data presented in this study are available on request from the corresponding author.

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
