# Peer review of "Reaching a Wider Audience: Instagram’s Role in Dairy Cow Nutrition Education and Engagement"

_animals, 2023, doi:10.3390/ani13223503_

Round 1
Reviewer 1 Report
Comments and Suggestions for Authors
Thank you for this interesting paper. Although it is a communication paper, I believe that it would be useful to provide more specific cases such as communication campaigns taht can be labelled as a good practices, engagement tips and succesful strategies that were implemented. I also recommend to provide more specific explanations for some sentences. For example, what could be a clearer strategy for setting clear objectives? what could be the basis for goal setting in such an instagram account? what is your recommendation on optimal frequency to share posts?
Author Response
Dear Reviewer,
Thank you very much for taking the time to review our manuscript. Please find the detailed responses in the attachment below and the corresponding revisions highlighted in red.

Reviewer 2 Report
Comments and Suggestions for Authors
1.The introduction currently does not refer to or acknowledge existing research or practices regarding using Instagram as an educational tool. Acknowledging previous studies would provide a more robust background for the authors' novel efforts.
2.The proposed outcomes of the Instagram initiative can be more clearly defined. AItis uncertain whether the main goal is to create an interactive learning platform, engage the public with accurate information, or a combination of the two. Clear and defined objectives have strengthened the introduction.
3.While the text cites the rising popularity of Instagram among medical students [4,5], it does not present precise evidence to support its effectiveness as an educational tool in veterinary education. More rigorous evidence, such as studies with comparable situations or preliminary results from using Instagram in this context, would strengthen this argument.
4.This piece acknowledges the necessity of certain strategic elements for successful Instagram integration within veterinary education, but it does not address how success would be measured. The inclusion of preestablished metrics or Key Performance Indicators (KPIs) to evaluate the effectiveness of an educational Instagram account would contribute to a more informative analysis.
5.Several objectives range from educating specific groups to countering misinformation and showcasing successful practices. While principled, having numerous objectives might dilute the page’s focus and efficacy. Prioritizing or streamlining objectives may lead to a more impactful page.
6.Despite the enumeration of objectives, there is no mention of how to evaluate the achievement of these objectives. There is a need for specific metrics or indicators that can quantitatively or qualitatively measure the outcomes of stated objectives.
7.The description of the posting content does not sufficiently emphasize user engagement. Strategies for ensuring interactive content, such as user-generated content or inviting users to participate in conversations, can be integrated to foster higher levels of engagement and learning.
8.Instagram's strength lies in visual and multimedia content, but text does not adequately address this aspect. Further discussion on how photos, videos, or other types of rich media are used to deliver educational content would provide a more complete understanding of the strategy employed for this Instagram page.
9.The authors note that they adhere to the platform's image and video quality guidelines but do not outline any internal quality control measures for their posts. It would be useful for them to discuss methods employed to ensure the quality of content, including fact-checking and peer review processes.
10.Although the authors described using various features, such as stories, hashtags, reels, and live videos, they could further elaborate on how exactly these features are used strategically to boost engagement and achieve their stated goals. Specific examples can help clarify this.
11.The authors state how they have used Instagram to target different learning styles and interests, and to reach out to both the general public and specialists. However, they might strengthen their case by providing some examples of measures or metrics they are using to assess the effectiveness of these methods for different types of content and audiences.
12.Any study should identify the challenges or limitations of this study. Providing introspection and identifying areas of possible improvement could strengthen the overall depth and value of this study.
13.While the conclusion briefly mentions future possibilities, it could be constructive to include specific ways they intend to adapt their strategy to the ever-evolving features of social media platforms or ideas for future research in this area.
Author Response
Dear reviewer 2,
thank you for your comments,
please see the attached file
Best regards

Reviewer 3 Report
Comments and Suggestions for Authors
Regardless of the many goals of the study/article that were given in the initial part of the article, it would be worth writing what was the cognitive (scientific) goal and what was the utilitarian (practical) goal of the considerations undertaken. Or was it just a practical purpose? You can also write about the gaps in the current state of knowledge (the use of social networking sites to promote specialized, scientific agricultural knowledge?), which prompted the authors to take up the topic and present it in the article. On this basis, it is possible to indicate a gap in the current state of knowledge / approach to research involving the use of the specificity of the mass media to develop and disseminate specialized agricultural knowledge.
Is it fully justified to include access to scientific articles for Instagram users, i.e. farmers? Was/is there a selection of scientific articles shared on Instagram with farmers? In my opinion, some scientific articles present very specialized knowledge in a narrow scope and it is difficult to directly translate the knowledge into practical solutions for use on the farm. Hence, another question arises: Are popular science articles that present scientific knowledge in a slightly different form, accessible to farmers, also included on Instagram?
On line 18 I don't know what "caliber of dairy products" is. I'm not sure the word "caliber" is used correctly here.
I think that a valuable element of the presented study on the use of Instagram to promote and popularize knowledge in the area of dairy cattle production, their nutrition and welfare would be to include selected opinions of users of the social media in question. It would be valuable to present the opinions of students, farmers and other social groups using a given (considered) social networking site. The article could also present the actual effects of educating students using Instagram.
I would like to ask how the effects of student education are verified using the tool in question, which is Instagram? Are these some tests or other tools, such as in MS Teams? It's worth writing about it.
In the article you can write something about the prospects for the development of education using Instagram. In the final part of the article, it would also be worth writing and discussing the factors limiting the presented teaching method and its application in practice also in other academic and other communities.
Author Response
Dear Reviewer 3,
thank you for your comments
please see the attached file

Round 2
Reviewer 2 Report
Comments and Suggestions for Authors
The text has been revised by incorporating feedback provided by the reviewer. The authors have addressed my comments and suggestions, thereby enhancing both the quality and clarity of the manuscript. I am now in a position to recommend acceptance.